# LANGUAGE MODELS ARE HIDDEN REASONERS: UNLOCKING LATENT REASONING CAPABILITIES VIA SELF-REWARDING

## ABSTRACT

Large language models (LLMs) have shown impressive capabilities, but still struggle with complex reasoning tasks requiring multiple steps. While prompt-based methods like Chain-of-Thought (CoT) can improve LLM reasoning at inference time, optimizing reasoning capabilities during training remains challenging. We introduce **LaT**ent **R**easoning **O**ptimization (**LaTRO**), a principled framework that formulates reasoning as sampling from a latent distribution and optimizes it via variational approaches. LaTRO enables LLMs to concurrently improve both their reasoning process and ability to evaluate reasoning quality, without requiring external feedback or reward models. We validate LaTRO through experiments on GSM8K and ARC-Challenge datasets using multiple model architectures. On GSM8K, LaTRO improves zero-shot accuracy by an average of 12.5% over base models and 9.6% over supervised fine-tuning across Phi-3.5-mini, Mistral-7B, and Llama-3.1-8B. Our findings suggest that pre-trained LLMs possess latent reasoning capabilities that can be unlocked and enhanced through our proposed optimization approach in a self-improvement manner.

## 1 INTRODUCTION

The development of large language models (LLMs) with enhanced reasoning capabilities has emerged as a crucial area of research. Despite their impressive advances, the inherent next-token prediction mechanism of LLMs makes it challenging for these models to solve complex problems requiring multiple reasoning steps (Wang et al., 2022; Huang et al., 2023). For instance, LLMs often struggle to directly provide accurate solutions to mathematical problems or even simple puzzles like counting specific letters in a word. Consequently, researchers have explored various prompting strategies that guide LLMs to generate reasoning trajectories or rationales—sequences of tokens that build a step-by-step progression toward an answer. Techniques such as Chain-of-Thought (CoT) (Wei et al., 2022), Tree-of-Thought (ToT) (Yao et al., 2024), and Program-of-Thought (PoT) (Chen et al., 2023) prompting methods exemplify this approach.

Recent progress has also focused on inference-time techniques to enhance the reasoning abilities of LLMs (Wu et al., 2024; Brown et al., 2024), as observed in the OpenAI o1 model (OpenAI, 2024). These methods have demonstrated remarkable performance in diverse reasoning tasks, including mathematics (Cobbe et al., 2021b; Trinh et al., 2024; Luo et al., 2024), coding (Jimenez et al., 2023; Guo et al., 2024; Zhang et al., 2024), and scientific problem-solving (Rein et al., 2023). Notable inference-time methods, such as CoT with Self-Consistency (CoT-SC) (Wang et al., 2023) and CoT-Decoding (Wang & Zhou, 2024), extend the CoT approach by generating multiple reasoning paths and selecting the most consistent one. Additionally, techniques like ReAct (Yao et al., 2023a) and Reflexion (Shinn et al., 2023) integrate reasoning into LLM agent loops, further enhancing their problem-solving capabilities.

Despite the promising results at inference time, improving the reasoning abilities of LLMs during their training phase remains a challenging problem. Several obstacles impede progress in this area. Firstly, there is a scarcity of high-quality reasoning data for complex problems, limiting the applicability of traditional supervised fine-tuning (SFT) approaches (Zelikman et al., 2022). Moreover, when such data is available, SFT on deterministic reasoning paths may result in a lack of diversity in

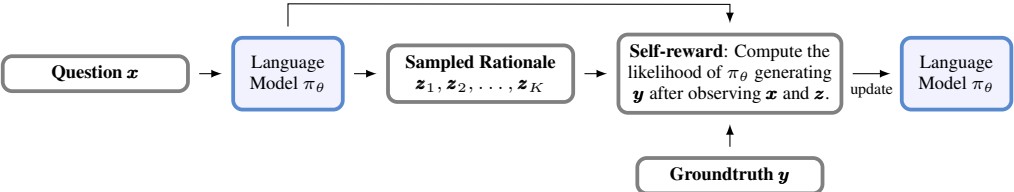

**Question**: A robe takes 2 bolts of blue fiber and half that much white fiber. How many bolts does it take?
**Groundtruth**: The answer is 3.
**Sampled Rationale 1 (correct ✅, higher likelihood)**: It takes 2/2 = 1 bolt of white fiber. 2 + 1 = 3. So, it takes a total of 3 bolts of fiber.
**Sampled Rationale 2 (incorrect ❌, lower likelihood)**: We need 2 bolts of blue and 2 bolts of white fiber. In total, it is 2 + 2 = 4.

Figure 1: Overview of LaTRO with an example question from GSM8K (Cobbe et al., 2021b). LaTRO treats reasoning trajectories as latent variables and optimizes the underlying distribution through self-rewarding. Given a question, the language model generates multiple reasoning rationales, evaluates their likelihood of producing the correct answer, and updates its parameters to favor high-quality rationales. This iterative process allows the model to improve both its ability to generate good reasoning paths and to evaluate the quality of those paths.

problem-solving strategies, potentially causing over-confidence issues and performance degradation (Cobbe et al., 2021b), especially in domains needing multiple valid approaches, such as mathematical proofs and coding. Alternatively, improving reasoning through reinforcement learning from human feedback (RLHF) presents its own challenges (Havrilla et al., 2024; Luo et al., 2024). Developing a reward model that accurately evaluates the quality and validity of reasoning paths is a formidable task, susceptible to distribution shifts and biased evaluations.

Self-improvement approaches like STaR (Self-Taught Reasoner) (Zelikman et al., 2022) and Quiet-STaR (Zelikman et al., 2024) have shown promise in enhancing language models' reasoning capabilities without external feedback. However, STaR relies on task-specific few-shot examples to bootstrap its reasoning process, which can limit its generalizability across diverse tasks. While Quiet-STaR attempts to overcome this by inferring implicit rationales across arbitrary text, it does not directly optimize the reasoning process itself. Through these findings, we observe that *pretrained LLMs already possess innate reasoning capabilities but just have not been fully activated or utilized*, inspiring us to propose our approach.

Our proposed method, **LaT**ent **R**easoning **O**ptimization (**LaTRO**), addresses the limitations of previous approaches by formulating reasoning as sampling from a latent distribution and optimizing it through a principled variational framework. As illustrated in Fig. 1, LaTRO enables language models to concurrently improve both their reasoning process and ability to evaluate reasoning quality, without requiring task-specific few-shot examples or external reward models. Key contributions of LaTRO include:

1. A theoretical formulation connecting LLM reasoning optimization to latent variable models;

2. A self-rewarding mechanism leveraging the model's own probability estimates;

3. Significant performance gains across multiple model architectures and reasoning tasks, demonstrating LaTRO's effectiveness in unlocking latent reasoning capabilities of language models.

Our findings suggest that pre-trained LLMs are not only capable reasoners but also possess the potential to act as explicit reward models for evaluating reasoning paths. We term this approach of utilizing explicit reward functions induced by LLMs themselves as "self-rewarding." Empirically, LaTRO outperforms both baseline models and supervised fine-tuning approaches on reasoning tasks like GSM8K, while also demonstrating the capacity to compress reasoning processes and shift computational burdens from inference to training time.

## 2 RELATED WORK

**Prompt-based LLM Reasoning**   Prompt-based reasoning methods prove to be effective across various domains, such as math problem-solving (Polu & Sutskever, 2020; Hendrycks et al., 2021; Cobbe et al., 2021a), logical reasoning (Sprague et al., 2024) and agentic tasks (Yao et al., 2023a; Shinn et al., 2023; Yao et al., 2023b). Chain-of-Thoughts or CoT (Wei et al., 2022) is the pioneering work that prompts LLMs to decompose challenging tasks into smaller reasoning steps. After that, two primary research directions further improved reasoning capabilities during inference. One direction searched over the reasoning trajectories against a process-based verifier, or reward model (Yao et al., 2024; Besta et al., 2024; Lightman et al., 2023). For example, tree-of-thoughts (Yao et al., 2024) explored over thoughts by depth-first search (DFS), breadth-first search (BFS) or beam search. The other approach used a critic model to provide verbal feedback, iteratively refining the responses with that feedback (Saunders et al., 2022; Shinn et al., 2023; Yao et al., 2023b; Madaan et al., 2023).

**Self-Rewarding for LLM Reasoning**   Reasoning capabilities in LLMs can be enhanced in post-training through self-rewarding and reinforcement learning. The Self-Taught Reasoner, or STaR (Zelikman et al., 2022) introduced a bootstrapping technique that allows LLMs to generate rationales and fine-tune itself with self-generated reasoning paths. Quiet-STaR (Zelikman et al., 2024) extended this by training LLMs to infer implicit rationales across arbitrary text, enhancing both reasoning and predictive abilities without task-specific fine-tuning. Reinforced Fine-Tuning, or ReFT (Trung et al., 2024) took this further by leveraging reinforcement learning to improve generalization in reasoning tasks like math problem-solving, enabling LLMs to learn from multiple reasoning paths. Self-correction capabilities in LLMs can also be reinforced through self-generated data (Kumar et al., 2024). Lastly, Hoffman et al. (2024); Hu et al. (2023) formulated the reasoning process as latent variable models, aligning LLMs towards more accurate reasoning with fewer annotated data.

## 3 BACKGROUND AND MOTIVATION

We start by briefly introducing reasoning techniques (*e.g.*, chain-of-thought (Wei et al., 2022), ReAct (Yao et al., 2023a), *etc*). Given a user query $\boldsymbol{x}$, the standard procedure to sample the response $\boldsymbol{y}$ is to leverage an autoregressive pretrained LLMs $\pi_\theta$ (parameterized by $\theta$): $\boldsymbol{y} \sim \pi_\theta(\cdot \mid \boldsymbol{x})$. As for prompt-based reasoning techniques such as chain-of-thought (Wei et al., 2022), the LLM $\pi_\theta$ is firstly asked to generate thoughts (*a.k.a* reasoning rationale) before generating the answers to the response:

$$\boldsymbol{x}' := \texttt{Reason}(\boldsymbol{x}) = \boldsymbol{x} \oplus \texttt{Prompt Template of Thought},$$

$$\boldsymbol{z} \sim \pi_\theta(\cdot \mid \boldsymbol{x}'), \quad \boldsymbol{y} \sim \pi_\theta(\cdot \mid \boldsymbol{x}' \oplus \boldsymbol{z}),$$

where $\boldsymbol{z}$ is the thought or the reasoning rationale path, $\oplus$ indicates the concatenate operator, and the prompt template of the thought can be some hint prompt such as "Let's think step by step" [1].Empirically, people observe that there is a higher chance for the LLM $\pi_\theta$ to generate the desired answer $\boldsymbol{y}$ following the above procedure than directly sampling the response $\boldsymbol{y} \sim \pi_\theta(\cdot \mid \boldsymbol{x})$. From a statistical perspective, we hypothesize that good reasoning rationales can significantly improve the probability of generating good answers $\boldsymbol{y}$: $\exists \, \boldsymbol{z}, \quad s.t. \; \pi_\theta(\boldsymbol{y}\mid \boldsymbol{x} \oplus \boldsymbol{z}) \gg \pi_\theta(\boldsymbol{y}\mid \boldsymbol{x})$.

To validate the hypothesis, we check the probability of the correct answers $\boldsymbol{y}$ on pretrained LLMs with or without reasoning rationales. In Figure 2, we visualize the negative log probability of the correct answers on three different LLMs on GSM8K dataset (Cobbe et al., 2021b). We have observed that when the LLMs are conditioned on the reasoning rationales, the probability of the correct answer is remarkably larger than without reasoning rationales. This suggests that good reasoning rationales help the LLMs generate desired answers for complex tasks by increasing their probability, giving them a higher chance of generating the correct answers.

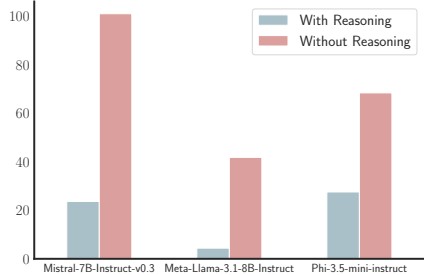

Figure 2: Average negative log probabilities of LLMs to generate correct responses.

---

[1]We omit the difference between $x'$ and $x$ for convenience in the latter notation.

The above observation inspires us that we can potentially further improve the reasoning ability of existing LLMs. One may find some surrogate objective to enhance the quality of the reasoning rationales or improve the ability of LLMs to leverage good reasoning rationales. In the following Proposition 1, we show that *Self-Consistency* Chain-of-Thought (CoT-SC) (Wang et al., 2023), which takes a majority vote of multiple reasoning rationales to improve the reasoning ability, approximates some surrogate objective.

**Proposition 1.** *Denote the user query, model response, and reasoning rationale by $\boldsymbol{x}, \boldsymbol{y}, \boldsymbol{z}$, respectively. The distribution of the majority vote answer of the $K$ reasoning rationales obtained by CoT-SC approximates $p_M(\boldsymbol{y}|\boldsymbol{x}) := \mathbb{E}_{\boldsymbol{z} \sim \pi_\theta(\cdot \mid \boldsymbol{x})}[\pi_\theta(\boldsymbol{y}|\boldsymbol{x} \oplus \boldsymbol{z})]$, as $K \to \infty$.*

*Proof.* Given a user query $\boldsymbol{x}$, CoT-SC essentially follows the procedure: 1) Sample $K$ *i.i.d* reasoning rationales together with model responses: $(\boldsymbol{z}_k, \boldsymbol{y}_k) \sim \pi_\theta(\cdot|\boldsymbol{x})$, $1 \leq k \leq K$. 2) Take the majority vote of $(\boldsymbol{y}_1, \ldots, \boldsymbol{y}_K)$. For a specific response $\boldsymbol{y}$, its frequency can be calculated as $F(\boldsymbol{y}) := \frac{1}{K} \sum_{k=1}^{K} \mathbb{1}\{\boldsymbol{y}_k = \boldsymbol{y}\}$, where $\mathbb{1}$ is the indicator function. Then the expectation of $F(\boldsymbol{y})$ is

$$\mathbb{E}_{\boldsymbol{y}_1, \ldots, \boldsymbol{y}_K} F(\boldsymbol{y}) = \frac{1}{K} \sum_{k=1}^{K} \mathbb{E}_{\boldsymbol{y}_i} \mathbb{1}\{\boldsymbol{y}_i = y\} = \frac{1}{K} \sum_{i=1}^{K} \mathbb{P}_{\boldsymbol{y}_i \sim \pi_\theta(\cdot|\boldsymbol{x} \oplus \boldsymbol{z}_i)}[\boldsymbol{y}_i = \boldsymbol{y}]$$

$$= \frac{1}{K} \sum_{i=1}^{K} \pi_\theta(\boldsymbol{y}|\boldsymbol{x} \oplus \boldsymbol{z}_i) \overset{K \to \infty}{\longrightarrow} \mathbb{E}_{\boldsymbol{z} \sim \pi_\theta(\cdot|\boldsymbol{x})} \pi_\theta(\boldsymbol{y}|\boldsymbol{x} \oplus \boldsymbol{z}).$$

$\square$

CoT-SC essentially leverages $p_M(\boldsymbol{y}|\boldsymbol{x}) := \int \pi_\theta(\boldsymbol{z}|\boldsymbol{x}) \pi_\theta(\cdot|\boldsymbol{x} \oplus \boldsymbol{z}) d\boldsymbol{z}$ to obtain reasoning rationales and produces final correct answers. Inspired by the conclusion, we could leverage surrogate objectives like $\mathbb{E}_{\boldsymbol{z} \sim \pi_\theta(\cdot \mid \boldsymbol{x})}[\phi(\pi_\theta(\boldsymbol{y}|\boldsymbol{x} \oplus \boldsymbol{z}))]$ to further enhance the reasoning ability of LLMs, where $\phi$ is some monotonic transform such as logarithm ($\log(\cdot)$). Further, we could also optimize the parameters of LLMs to enhance the reasoning abilities of existing LLMs during the training, so we can obtain LLMs with better reasoning abilities with the same inference time budget. In the following sections, we introduce the idea of optimizing LLMs to improve reasoning abilities without external feedback, by proposing a principled framework containing the surrogate objective.

## 4 OPTIMIZING THE REASONING PROCESS

In this section, we describe how to optimize the reasoning rationale without external feedback. Specifically, we introduce the objective for optimizing the reasoning rationale in Section 4.1 from a variational perspective of LLM training; we derive the gradient estimation for the new objective in Section 4.2, and discuss the sampling procedure together with reward shaping in Section 4.3. We summarize the proposed algorithm, **LaT**ent **R**easoning **O**ptimization (**LaTRO**) in Algorithm 1, and illustrate the overall procedure in Figure 1.

### 4.1 LATENT REASONING OPTIMIZATION: A VARIATIONAL APPROACH

Suppose we are given a golden dataset $\mathcal{D}_{\text{Gold}} := \{(\boldsymbol{x}_i, \boldsymbol{y}_i)\}_{i=1}^{N}$ consisting of $N$ query and answer pairs, where $(\boldsymbol{x}, \boldsymbol{y})$ denotes the query and the answer respectively. A standard finetuning procedure to fit the LLM $\pi_\theta$ to the dataset $\mathcal{D}_{\text{Gold}}$ is by likelihood maximization:

$$\max_\theta \mathbb{E}_{(\boldsymbol{x}, \boldsymbol{y}) \sim \mathcal{D}_{\text{Gold}}} [\log \pi_\theta(\boldsymbol{y} \mid \boldsymbol{x})], \tag{1}$$

where $\theta$ are the parameters of the LLM $\pi_\theta$ to optimize. Based on the discussion in Section 3, it is more feasible to optimize $\pi_\theta$ with additional reasoning rationale path $\boldsymbol{z}$, compared with standard finetuning objective in Equation (1). Hence, we can introduce another "reasoner" $q(\boldsymbol{z}|\boldsymbol{x})$ to sample the latent reasoning rationales that can help the optimization procedure of $\pi_\theta$. This is achievable by

optimizing the following lower bound:

$$\log \pi_\theta(\boldsymbol{y}|\boldsymbol{x}) = \log \int \pi_\theta(\boldsymbol{y} \mid \boldsymbol{x} \oplus \boldsymbol{z}) \pi_0(\boldsymbol{z} \mid \boldsymbol{x}) d\boldsymbol{z}$$

$$= \log \int \pi_\theta(\boldsymbol{y} \mid \boldsymbol{x} \oplus \boldsymbol{z}) \frac{q(\boldsymbol{z}|\boldsymbol{x})}{q(\boldsymbol{z}|\boldsymbol{x})} \pi_0(\boldsymbol{z}|\boldsymbol{x}) d\boldsymbol{z}$$

$$\geq \max_{q(\boldsymbol{z}|\boldsymbol{x})} \mathbb{E}_{q(\boldsymbol{z}|\boldsymbol{x})} \big[ \log \pi_\theta(\boldsymbol{y}| \boldsymbol{x} \oplus \boldsymbol{z}) \big] - D_{\mathrm{KL}}[q(\boldsymbol{z}|\boldsymbol{x})||\pi_0(\boldsymbol{z}|\boldsymbol{x})], \qquad (2)$$

where $\pi_0$ is a prior reference LLM that regularizes the "reasoner" $q(\boldsymbol{z}|\boldsymbol{x})$, and the lower bound is achieved via Jensen's inequality (Higgins et al., 2017). Based on the literature of variational Bayes (Kingma, 2013), one can either learn and optimize $q(\boldsymbol{z}|\boldsymbol{x})$ via variational Expectation Maximization (EM) (Abdolmaleki et al., 2018; Liu et al., 2022), or introduce another parameterized LLM $q_\phi(\boldsymbol{z}|\boldsymbol{x})$ and optimize $\phi$ to amortize the cost. Additionally, from the discussion in Section 3, we know $\pi_\theta$ itself can also serve as a naive "reasoner", since $\pi_\theta$ is an autoregressive LLM.

To simplify the learning procedure, we propose to use $\pi_\theta$ as the "reasoner" $q(\boldsymbol{z}|\boldsymbol{x})$. As a result, we can jointly learn one single LLM $\pi_\theta$, that is capable of generating good reasoning rationale together with providing correct answers given the query and its own generated reasoning rationale. To be more specific, we can define the learning objective as follows:

$$\max_\theta J(\theta) := \mathbb{E}_{(\boldsymbol{x},\boldsymbol{y})\sim\mathcal{D}_{\mathrm{Gold}}} \Big[ \mathbb{E}_{\boldsymbol{z}\sim\pi_\theta(\cdot|\boldsymbol{x})} \big[ \underbrace{\log \pi_\theta(\boldsymbol{y}| \boldsymbol{x} \oplus \boldsymbol{z})}_{R_\theta(\boldsymbol{z},\boldsymbol{y},\boldsymbol{x})} \big] - D_{\mathrm{KL}}[\pi_\theta(\boldsymbol{z}|\boldsymbol{x})||\pi_0(\boldsymbol{z}|\boldsymbol{x})] \Big], \qquad (3)$$

where we specify the reference LLM $\pi_0$ to be the original $\pi_\theta$ before the optimization. Furthermore, $\log \pi_\theta(\boldsymbol{y}| \boldsymbol{x} \oplus \boldsymbol{z})$ in Equation (3) can be viewed as the reward function $R_\theta(\boldsymbol{z},\boldsymbol{y},\boldsymbol{x})$ to evaluate the quality of the rationale $\boldsymbol{z}$ given the pair $(\boldsymbol{x},\boldsymbol{y})$, since the reasoning rationale $\boldsymbol{z}$ with higher likelihood $\log \pi_\theta(\boldsymbol{y}| \boldsymbol{x} \oplus \boldsymbol{z})$ indicates that it would provide a higher probability for the model to answer the question correctly.

**Remark** By substituting $\log \pi_\theta(\boldsymbol{y}| \boldsymbol{x} \oplus \boldsymbol{z})$ with $R_\theta(\boldsymbol{z},\boldsymbol{y},\boldsymbol{x})$, Equation (3) exactly recovers the standard optimization objective defined in offline RL (Levine et al., 2020), RLHF (Ouyang et al., 2022; Rafailov et al., 2024) literature. Though Equation (3) unifies the learning procedure of the "reasoner" $\pi_\theta(\boldsymbol{z}|\boldsymbol{x})$ and the "reward" function $R_\theta(\boldsymbol{z},\boldsymbol{y},\boldsymbol{x}) := \log \pi_\theta(\boldsymbol{y}| \boldsymbol{x} \oplus \boldsymbol{z})$, we can break down these two procedures to analyze them separately. When we fix $R_\theta(\boldsymbol{z},\boldsymbol{y},\boldsymbol{x})$ and optimize the "reasoner" $\pi_\theta(\boldsymbol{z}|\boldsymbol{x})$, the procedure can be interpreted as *self-improvement* learning, where we improve $\pi_\theta(\boldsymbol{z}|\boldsymbol{x})$ on self-generated synthetic reasoning rationale. When we fix $\pi_\theta(\boldsymbol{z}|\boldsymbol{x})$ and optimize $R_\theta(\boldsymbol{z},\boldsymbol{y},\boldsymbol{x})$, the procedure can be interpreted as *self-reward* learning, where we learn the self-reward function $\log \pi_\theta(\boldsymbol{y}| \boldsymbol{x} \oplus \boldsymbol{z})$. The procedure can also be considered finetuning optimization given the learned reasoning rationale and query. Fortunately, we can naturally enjoy the benefits of these two self-learning procedures with the new reasoning finetuning objective.

### 4.2 GRADIENT ESTIMATION FOR LaTRO

From previous RL literature, we know that estimating $\nabla_\theta J(\theta)$ in Equation (3) involves the use of policy gradient methods, which usually suffers from high variances with the naive REINFORCE estimators (Williams, 1992). Inspired by the recent work on policy gradient for LLMs (Ahmadian et al., 2024), we also leverage the REINFORCE Leave-One-Out (RLOO) (Kool et al., 2019) to optimize the "reasoner" $\pi_\theta(\boldsymbol{z}|\boldsymbol{x})$, where we can achieve lower variances of gradient estimation by sampling multiple rationales. We summarize the empirical gradient estimation for solving LaTRO in Proposition 2.

**Proposition 2.** *(LaTRO Gradient Estimation) Suppose we are given a set of training data $\mathcal{D}_{Gold} := \{\boldsymbol{x}_i, \boldsymbol{y}_i\}_{i=1}^N$, we sample $K$ i.i.d reasoning rationales $\boldsymbol{z}_1^{(i)}, \boldsymbol{z}_2^{(i)}, \ldots, \boldsymbol{z}_K^{(i)} \sim \pi_\theta(\cdot|\boldsymbol{x}_i)$ for each query and answer pair $(\boldsymbol{x}_i, \boldsymbol{y}_i)$. The empirical gradient estimator for $\nabla_\theta J(\theta)$ is expressed as*

$$\nabla_\theta \widehat{J}(\theta) := \frac{1}{NK} \sum_{i=1}^N \sum_{k=1}^K \left( \nabla_\theta \log \pi_\theta(\boldsymbol{x}_i \mid \boldsymbol{z}_k^{(i)}) \cdot A_k^{(i)} + \nabla_\theta \log \pi_\theta(\boldsymbol{y}_i \mid \boldsymbol{z}_k^{(i)} \oplus \boldsymbol{x}_i) \right), \qquad (4)$$

*with* $A_k^{(i)} = r(\boldsymbol{z}_k^{(i)}) - \frac{1}{K-1} \sum_{j\neq k}^K r(\boldsymbol{z}_j^{(i)}), r(\boldsymbol{z}_k^{(i)}) := \log \pi_\theta(\boldsymbol{y}_i \mid \boldsymbol{x}_i \oplus \boldsymbol{z}_k^{(i)}) - \beta \log \frac{\pi_\theta(\boldsymbol{x}_i \mid \boldsymbol{z}_k^{(i)})}{\pi_0(\boldsymbol{x}_i \mid \boldsymbol{z}_k^{(i)})},$

---

**Algorithm 1:** **LaT**ent **R**easoning **O**ptimization (**LaTRO**)

---

**Input:** Language model $\pi_\theta$, learning rate $\eta$, KL penalty factor $\beta$, MC sample size $K$, maximum generation length $L$, sample temperature $T$, number of epochs $M$, training dataset $\mathcal{D}_{\text{Gold}}$.

**Output:** An optimized language model $\pi_\theta$.

---

**def** generate($\pi, \boldsymbol{x}, K, L, T$)**:**
   | Given an autoregressive language model $\pi$, input $x$, sample $K$ sequences of length $L$ from
   | the distribution $\pi(\cdot|\boldsymbol{x})$ at temperature $T$.
   | **return** $K$ sampled sequences
Intialize reference language model $\pi_0$ as $\pi_\theta$
**for** *epoch* **in** range($M$)**:**
   | **for** $\boldsymbol{x}_i, \boldsymbol{y}_i$ **in** $\mathcal{D}_{Gold}$**:**
   |    | $\boldsymbol{z}_1^{(i)}, \ldots, \boldsymbol{z}_K^{(i)} \leftarrow$ generate($\pi_\theta, \boldsymbol{x}_i, K, L, T$)
   | Estimate $\nabla_\theta \widehat{J}(\theta)$ with Proposition 2
   | $\theta \leftarrow \theta + \eta \nabla_\theta \widehat{J}(\theta)$
**return** $\pi_\theta$

---

*where $\beta \geq 0$ is the coefficient to control the KL penalty. The proof can be found in Appendix A.1.*

The first gradient term in Equation (4) serves as policy gradient to improve the ability of the LLM $\pi_\theta$ to generate high-quality reasoning rationales, and $\log \pi_\theta(\boldsymbol{y}|\boldsymbol{x} \oplus \boldsymbol{z})$ can be viewed as the evaluator for reasoning rationale, which is further used to calculate the advantages. The second gradient term in Equation (4), which is the gradient of supervised finetuning loss, essentially helps the LLM $\pi_\theta$ to leverage the reasoning rationales to produce correct answers.

### 4.3 PRACTICAL CONSIDERATIONS

To reduce computation overhead and better control the sampling of reasoning rationales during training, we limit their maximum token length to $L$. The rationale ends either at the EOS token or at the start of a predefined answer template (e.g., "The answer is"). We then use the truncated rationale $\boldsymbol{z}$, along with the query $\boldsymbol{x}$ and the answer $\boldsymbol{z}$, for further computation.

We also encourage the LLM to finish its reasoning process with $L$ tokens. Inspired by the implementation of the RLOO trainer in the TRL library (von Werra et al., 2020) , we introduce a constant penalty for rationales truncated by the maximum token length $L$. This penalty encourages the generation of rationales that fit within the specified token limit.

## 5 EXPERIMENTS

### 5.1 SETUP

We evaluate the performance of the proposed method across two datasets: a mathematical reasoning dataset (GSM8K, Cobbe et al. (2021b)) and a logical reasoning dataset (ARC-Challenge, Talmor et al. (2019)). The sizes of the datasets are listed in Table 1.

**Training.** For each dataset, we fine-tune three base models: Phi-3.5-mini-instruct (Abdin et al., 2024), Mistral-7B-Instruct-v0.3 (Jiang et al., 2023), and Meta-Llama-3.1-8B-Instruct (Dubey et al., 2024), abbreviated as Phi-3.5, Mistral-7B, and Llama-3.1-8B, respectively. We provide two baseline comparisons: the base model and the super-

Table 1: Size of the datasets

| Name | Training | Evaluation |
|---|---|---|
| GSM8K | 7473 | 1319 |
| ARC-Challenge | 1119 | 1172 |

vised fine-tuned (SFT) model. For GSM8K, LaTRO fine-tuning excludes golden rationales from the solutions in the training set, while the SFT model is trained using golden rationales. For ARC-Challenge, as suggested in (Zheng et al., 2024), the model is trained to generate answers to the text of

multiple-choice questions rather than selecting labels. Since no golden rationales are available for ARC-Challenge, the SFT model is trained to directly generate answers.

**Evaluation.** For GSM8K, we evaluate all models with CoT prompting, and for ARC-Challenge, we evaluate the SFT baseline with direct answer generation, while the base model and the LaTRO fine-tuned model with CoT prompting. All evaluations are conducted with zero-shot prompts. We report both greedy decoding (GD) results and self-consistency (with temperature $T = 1$) results. We choose a self-consistency sample size $k = 8$ (maj@8) in Table 2 after observing that more than 8 samples did not bring further performance improvement (see Figure 3 (b) for details).

**Implementation.** LaTRO is implemented on the high level as in Algorithm 1, with additional engineering techniques as discussed in section 4.3. LaTRO is implemented using the widely recognized transformers (Wolf et al., 2020) and TRL (von Werra et al., 2020) libraries, with PyTorch (Ansel et al., 2024) as backend. DeepSpeed ZeRO (Rasley et al., 2020) is used in stage 3, along with Flash Attention 2 (Dao et al., 2022) to enhance training efficiency. The models were trained on a machine equipped with 8xH100 80GB GPUs, using bfloat16 precision.

**Hyperparameters.** AdamW optimizer with a learning rate of $5 \times 10^{-7}$, no warm-up steps, and a linear decay strategy is used. The Monte Carlo (MC) sample size $K = 16$ and the batch size of the data loader 3 are predetermined, resulting in an effective batch size of $48$. Gradient accumulation steps and training batch size are subsequently adjusted to prevent out-of-memory errors during training. A temperature of $T = 1$ is used for MC sampling, and a penalty factor $\gamma = 2$ is applied for incomplete rationales. The KL penalty is set at $\beta = 0.05$ for GSM8K and 0.25 for ARC-Challenge. Except for the results presented in Section 5.3, the maximum generation length is maintained at $L = 500$. We train all models up to six epochs for GSM8K, and 12 epochs for ARC-Challenge. The checkpoint with best test accuracy is chosen.

For the SFT baseline experiments, we use a batch size of 32 and adjust the learning rate to ensure that the evaluation loss decreases and finally converges. All SFT baselines are trained for a maximum of 12 epochs. The checkpoint with the best test accuracy is selected.

In addition to the main quantitative results, we conduct ablation studies on two factors: 1. The maximum generation length $L$, where we study the effects of tuning $L$ in both training and inference times; 2. The self-consistency samples $k$, where we explore to what extent LaTRO can still benefit from inference-time scaling.

The main quantitative results, qualitative analysis of sample responses, and results of the ablation study are presented in Sections 5.2 to 5.4, respectively. Additional details on our prompt templates and more samples can be found in Appendices B and C.

## 5.2 RESULTS

In this subsection, we present evaluation results that demonstrate how effectively LaTRO enhances the reasoning abilities of LLMs on downstream datasets. The detailed results are provided inTable 2.

For the GSM8K dataset, LaTRO fine-tuned models outperform all base models by up to $19.5\%$ (Mistral-7B, $47.8\% \to 67.3\%$) and show an average improvement of $12.5\%$ across the three models examined with greedy decoding. The greatest improvement margin is observed for Mistral-7B, while the smallest is seen for Llama-3.1-8B, consistent with our initial findings in Figure 2, where Mistral-7B exhibited the lowest log probability for directly answering questions and Llama-3.1-8B exhibited the highest. With self-consistency, the improvements are by up to $16.5\%$ (Phi-3.5, $74.0\% \to 90.5\%$) and the average improvement is $13.1\%$. Furthermore, LaTRO models demonstrate superior performance relative to SFT baselines, with an average improvement of $9.6\%$ for greedy decoding and $13.2\%$ for self-consistency. It is worth noting that for the SFT baseline of Llama-3.1-8B, overfitting on the test set is still observed after tuning the learning rate.

For ARC-Challenge, LaTRO fine-tuned models still outperform the baselines, though with a smaller margin. When using greedy decoding, the improvements over the base models are up to $1.6\%$ with an average increase of $1\%$. We see more increment with self-consistency, where the improvement margins are on average $2.4\%$. Comparing to SFT baslines, we find that all three models are very sensitive when fine-tuning to directly generate the answer of ARC-Challenge questions. They perform

Table 2: Zero-shot accuracy (%) comparison between LaTRO and the baselines on GSM8K and ARC-Challenge datasets. The models are fine-tuned on corresponding training datasets. The base model are marked with "N/A" in the training method. GD stands for greedy decoding at inference time and maj@8 stands for self-consistency with 8 samples. The models are evaluated by default using CoT, except that † indicates the direct answer generation is applied during evaluation.

| Base Model | Training Method | Inference Method | GSM8K | ARC-Challenge |
|---|---|---|---|---|
| Phi-3.5 | N/A | GD | 72.9 | 85.1 |
| | | maj@8 | 74.0 | 86.0 |
| | SFT | GD | 75.8 | 81.0† |
| | | maj@8 | 77.1 | 80.5† |
| | LaTRO | GD | **87.6** | **86.4** |
| | | maj@8 | **90.5** | **87.5** |
| Mistral-7B | N/A | GD | 47.8 | 74.1 |
| | | maj@8 | 58.2 | 74.1 |
| | SFT | GD | 57.2 | 70.0† |
| | | maj@8 | 59.9 | 70.6† |
| | LaTRO | GD | **67.3** | **74.3** |
| | | maj@8 | **73.8** | **78.9** |
| Llama-3.1-8B | N/A | GD | 76.8 | 81.4 |
| | | maj@8 | 79.7 | 84.4 |
| | SFT | GD | 73.2 | 77.0† |
| | | maj@8 | 74.7 | 76.4† |
| | LaTRO | GD | **80.1** | **83.0** |
| | | maj@8 | **87.0** | **85.3** |

even inferior to the unoptimized base models. When using greedy decoding, the improvements of LaTRO fine-tuned models over the SFT baselines are on an average of $5.2\%$, and by up to $6\%$ (Llama-3.1-8B). In the case of self-consistency, LaTRO performs better than the base models by an average of $2.4\%$, and surpasses the SFT models by an average of $8.1\%$. On the less surprising results compared to GSM8K, we conjecture that for ARC-Challenge, the models are already good at producing the answer either directly or through CoT prompting. Hence, further optimization of the reasoning process did not yield significant improvement.

## 5.3 ABLATION STUDY

In this subsection, we present our ablation study on the effect of different parameters in LaTRO. For consistency, we fix the base model to Phi-3.5 and the dataset to GSM8K throughout the ablation experiments.

**How many tokens are enough?** Liu et al. (2024) demonstrated that when the input length is $n$, a transformer model with a hidden size of $O(\log n)$ can solve problems equivalent to Boolean circuits of size $m$, using $m$ CoT steps. However, the empirical determination of sufficient CoT tokens for optimal performance remains underexplored. In this section, we report zero-shot accuracy with generation length $L$ ranging from 200 to 1000 tokens at inference time. Additionally, a Phi-3.5 model is fine-tuned with $L = 200$ for comparison. We distinguish two LaTRO fine-tuned models, referred to as LaTRO and LaTRO$_{200}$. As shown in Figure 3(a) accuracy gains plateau when $L \geq 500$, suggesting 500 tokens might suffice for grade school math problems. In contrast, limiting $L$ to 200 reduces accuracy, unless the model is trained accordingly. Interestingly, LaTRO significantly improves performance under this constraint by training the model to generate more concise rationales.

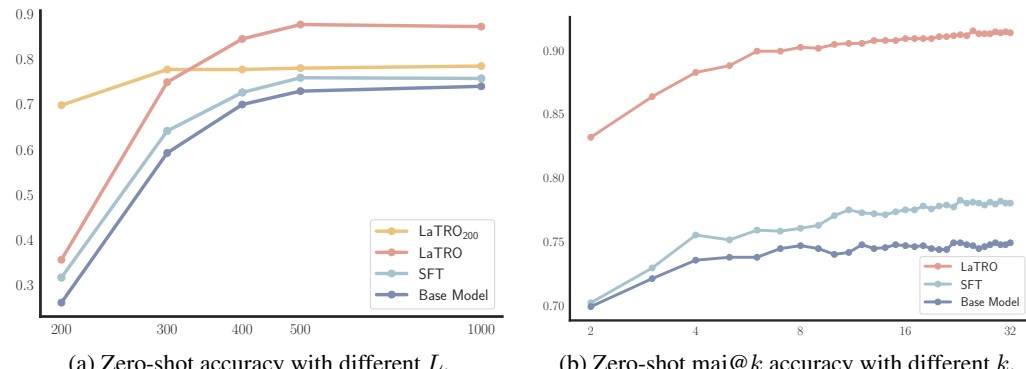

(a) Zero-shot accuracy with different $L$.  (b) Zero-shot maj@$k$ accuracy with different $k$.

Figure 3: Ablation study results on GSM8K with base model Phi-3.5. In (a), the $x$-axis represents various maximum token length $L$ of reasoning rationales, $y$-axis is the accuracy, and the plot shows the zero-shot performance v.s. various maximum token lengths for different methods. In (b), the $x$-axis represents the # of sampled reasoning rationales, the $y$-axis is the accuracy, and the plot shows the zero-shot performance v.s. the # of reasoning rationales used in the majority vote.

This finding implies LaTRO can also train LLMs to condense rationales for limited inference-time compute. A sample from LaTRO$_{200}$ is provided in Appendix C for qualitative analysis.

**Is inference-time scaling still beneficial?** While LaTRO shifts much of the reasoning computation to training time, it can still benefit from the inference-time scaling. To explore this, We evaluate the models again using self-consistency with the number of samples $k$ ranging from 1 to 32. As shown in Figure 3 (b), with $k = 8$ rationales and majority voting, LaTRO achieves 90% accuracy on GSM8K. This indicates that LaTRO equipped with self-consistency can further enhance performance, even with LaTRO's focus on training-time optimization.

> **Question**: Frankie and Binkie went bowling together. Frankie's score was 15 better more than twice as high as Binkie's. If Binkie bowled a score of 90, what was Frankie's score? Let's think step by step.
>
> **Response (Base model)**: Let's break this problem down into steps:
> 1. We know that the difference between Frankie's score (F) and Binkie's score (B) is 15. So we can express Frankie's score as F = B + 15
> 2. We also know that Frankie's score is twice as high as Binkie's score. So we can write another equation: F = 2B
> 3. Now, we have two equations with two unknowns. Let's solve for Frankie's score (F) by setting the two equations equal to each other: - 2B = B + 15 - Multiply both sides by 1 to get rid of the fraction: 2B - B = 15 - Simplify: B = 15 (since subtracting B from 2B results in 15)
> 4. Now that we know Binkie's score, we can find Frankie's score using either of the equations we created earlier. Let's use the first one (F = B + 15): - F = 15 + 15 - Frankie's score is 30 more than Binkie's score, which is 15 in this case. So Frankie's score is 15 + 15 = 30 + 15 = 45
> The final answer is that Frankie's score was 45.
>
> **Reponse (SFT)**: Twice as many as Binkie's 90 score is equal to 2*90= «2*90=180»180. Therefore, 180-(180-15)=165 was the number Frankie bowled. The answer is 165
>
> **Response (LaTRO)**: Here's how to solve the problem: 1. We know that Binkie's score is 90. 2. The problem states that Frankie's score was 15 better more than twice as high as Binkie's score. This means that Frankie's score is 15 + (2 * 90) = 15 + 180 = 195.

Figure 4: Sample responses of a GSM8K question from Mistral-7B models, the errors are highlighted in red.

### 5.4 Case Study

We take a closer look at the responses generated by the LaTRO fine-tuned models. We select a question from GSM8K and compare the responses from the base, the SFT model, and the LaTRO finetuned model. We choose the set of responses from the Mistral-7B models that we evaluated. As can be seen in Figure 4, the base model not only generates a lengthy response, it also makes a logical mistake at the first step, where the correct equation to establish here is "F = 2B + 15". The SFT model simplifies the answer and makes the first step correct. However, in the second step it first makes a wrong equation, then makes an arithmetic error when evaluating this equation. Further, LaTRO can give a concise and correct answer. We include more sample responses in Appendix C.

## 6 Conclusion

In conclusion, this work introduces LaTRO, a principled framework for optimizing language models' reasoning capabilities without external feedback or reward models. By formulating reasoning as sampling from a latent distribution and leveraging self-rewarding, LaTRO enables models to concurrently improve both their reasoning process and ability to evaluate reasoning quality. Our extensive experiments across multiple model architectures and tasks demonstrate significant performance gains, with LaTRO outperforming baseline models and supervised fine-tuning approaches. These findings suggest that pre-trained LLMs possess latent reasoning capabilities that can be unlocked through our proposed optimization approach, representing a significant step towards creating more intelligent systems that can self-evolve their problem-solving capabilities.

While LaTRO shows promising results, there are some limitations to consider. The computational cost of sampling multiple rationales during training could be prohibitive for very large models. Future work could explore ways to reduce this computational overhead, such as using more efficient sampling techniques or adaptive rationale generation. Other promising directions include investigating the applicability of LaTRO to a wider range of reasoning tasks beyond math and science problems, and exploring how to conduct multi-step reasoning learning to to enhance reasoning capabilities further. Despite these limitations, our contributions advance both the state-of-the-art in LLM reasoning capabilities and provide valuable insights into the nature of LLM alignment and its potential for self-improvement.

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

# A   ADDITIONAL DETAILS ON OUR THEORETICAL FRAMEWORK

## A.1   PROOF OF PROPOSITION 2

*Proof.* We restate the objective as follows:

$$J(\theta) := \mathbb{E}_{(\boldsymbol{x},\boldsymbol{y})\sim\mathcal{D}_{\text{Gold}}}\left[\mathbb{E}_{\boldsymbol{z}\sim\pi_\theta(\cdot|\boldsymbol{x})}\left[\log\pi_\theta(\boldsymbol{y}|\,\boldsymbol{x}\oplus\boldsymbol{z})\right] - \beta D_{\text{KL}}[\pi_\theta(\boldsymbol{z}|\boldsymbol{x})||\pi_0(\boldsymbol{z}|\boldsymbol{x})]\right],$$

$$= \mathbb{E}_{(\boldsymbol{x},\boldsymbol{y})\sim\mathcal{D}_{\text{Gold}}}\left[\mathbb{E}_{\pi_\theta(\boldsymbol{z}|\boldsymbol{x})}[\log\pi_\theta(\boldsymbol{y}|\boldsymbol{x}\oplus\boldsymbol{z}) - \log\pi_\theta(\boldsymbol{z}|\boldsymbol{x}) + \log\pi_0(\boldsymbol{z}|\boldsymbol{x})]\right],$$

where $\beta > 0$ is a positive coefficient to control the regularization strength. We take the gradient *w.r.t* $\theta$ at each sample pair $(\boldsymbol{x},\boldsymbol{y})$, and we get

$$\nabla_\theta J(\theta;\boldsymbol{x},\boldsymbol{y}) := \nabla_\theta \int (\pi_\theta(\boldsymbol{z}|\boldsymbol{x}))(\log\pi_\theta(\boldsymbol{y}|\boldsymbol{x}\oplus\boldsymbol{z}) - \beta\log\pi_\theta(\boldsymbol{z}|\boldsymbol{x}) + \beta\log\pi_0(\boldsymbol{z}|\boldsymbol{x}))d\boldsymbol{z}$$

$$= \mathbb{E}_{\pi_\theta(\boldsymbol{z}|\boldsymbol{x})}\left[\nabla_\theta\log\pi_\theta(\boldsymbol{z}|\boldsymbol{x})\left(\log\pi_\theta(\boldsymbol{y}|\boldsymbol{x}\oplus\boldsymbol{z}) - \beta\log\frac{\pi_\theta(\boldsymbol{z}|\boldsymbol{x})}{\pi_0(\boldsymbol{z}|\boldsymbol{x})}\right)\right]$$

$$+ \mathbb{E}_{\pi_\theta(\boldsymbol{z}|\boldsymbol{x})}\left[\nabla_\theta\log\pi_\theta(\boldsymbol{y}|\boldsymbol{x}\oplus\boldsymbol{z}) - \beta\nabla_\theta\log\pi_\theta(\boldsymbol{z}|\boldsymbol{x})\right].$$

We further define $r(\boldsymbol{z}) := \log\pi_\theta(\boldsymbol{y}|\boldsymbol{x}\oplus\boldsymbol{z}) - \beta\log\frac{\pi_\theta(\boldsymbol{x}|\boldsymbol{z})}{\pi_0(\boldsymbol{x}|\boldsymbol{z})}$, and use the fact that $\mathbb{E}_{\pi_\theta(\boldsymbol{z}|\boldsymbol{x})}[\nabla_\theta\log\pi_\theta(\boldsymbol{z}|\boldsymbol{x})] = \int\pi_\theta(\boldsymbol{z}|\boldsymbol{x})\frac{\nabla_\theta\pi_\theta(\boldsymbol{z}|\boldsymbol{x})}{\pi_\theta(\boldsymbol{z}|\boldsymbol{x})}d\boldsymbol{z} = \nabla_\theta\int\pi_\theta(\boldsymbol{z}|\boldsymbol{x})d\boldsymbol{z} = 0$. we obtain the final gradient as

$$\nabla_\theta J(\theta;\boldsymbol{x},\boldsymbol{y}) = \mathbb{E}_{\pi(\boldsymbol{z}|\boldsymbol{x})}\left[\nabla_\theta\log\pi_\theta(\boldsymbol{z}|\boldsymbol{x})\cdot r(\boldsymbol{z}) + \nabla_\theta\log\pi_\theta(\boldsymbol{y}|\boldsymbol{z},\boldsymbol{x})\right].$$

And when we use RLOO estimator with empirical samples, we can replace above gradient estimation with empirical samples, which gives us the following result:

$$\nabla_\theta\widehat{J}(\theta) := \frac{1}{NK}\sum_{i=1}^{N}\sum_{k=1}^{K}\left(\nabla_\theta\log\pi_\theta(\boldsymbol{x}_i\mid\boldsymbol{z}_k^{(i)})\cdot A_k^{(i)} + \nabla_\theta\log\pi_\theta(\boldsymbol{y}_i\mid\boldsymbol{z}_k^{(i)}\oplus\boldsymbol{x}_i)\right),$$

with $A_k^{(i)} = r(\boldsymbol{z}_k^{(i)}) - \frac{1}{K-1}\sum_{j\neq k}^{K}r(\boldsymbol{z}_j^{(i)}), r(\boldsymbol{z}_k^{(i)}) := \log\pi_\theta(\boldsymbol{y}_i\mid\boldsymbol{x}_i\oplus\boldsymbol{z}_k^{(i)}) - \beta\log\frac{\pi_\theta(\boldsymbol{x}_i\mid\boldsymbol{z}_k^{(i)})}{\pi_0(\boldsymbol{x}_i\mid\boldsymbol{z}_k^{(i)})}$,

which completes the proof. □

# B   PROMPT TEMPLATES

In this section, we list all prompt templates used in our experiments. For the GSM8K dataset, we used a minimum prompt template. For the ARC-Challenge dataset, we use a CoT template for the base model and LaTRO fine-tuning, and another template direct answer generation in SFT.

{question} Let's think step by step.

Figure 5: CoT template for GSM8K

Question: {question}
Options: {choices["text"]}
Let's think step by step first then print your answer like 'The answer is (one of the options)'.""",

Figure 6: CoT template for ARC-Challenge

The output MUST strictly follow the structure and format described below:
1. The answer should be strictly one of the options given in the question
2. Do not include any text after you choose an option.
**Example:**
Question: (question here)
Options: [option1, option2, option3]
The answer is: (one of the options).
Question: {question}
Options: {choices["text"]}

Figure 7: Direct answer generation template for ARC-Challenge

## C SAMPLE RESPONSES

**Question**: Kylar went to the store to buy glasses for his new apartment. One glass costs $5, but every second glass costs only 60% of the price. Kylar wants to buy 16 glasses. How much does he need to pay for them? Let's think step by step.
**Response (Base model)**:
Step 1: Determine the cost of one regular glass and one discounted glass. One regular glass costs $5. Every second glass costs 60% of the price of a regular glass, so: 60% of $5 = 0.60 * 5$ = $3
Step 2: Calculate the total number of regular and discounted glasses. Kylar wants to buy 16 glasses in total. Since every second glass is discounted, we can divide the total number of glasses by 2 to find the number of discounted glasses: 16 glasses / 2 = 8 discounted glasses Now, subtract the number of discounted glasses from the total number of glasses to find the number of regular glasses: 16 glasses - 8 discounted glasses = 8 regular glasses Step (did not finish)
**Response (LaTRO$_{200}$)**: Let's break it down:
1. Kylar needs to buy 16 glasses.
2. Every second glass costs 60% of the original price ($5), which is $3.
3. So, for every pair of glasses (one full price and one discounted), Kylar pays $5 + $3 = $8.
4. Since he needs 16 glasses, he buys 8 pairs. 5. Therefore, Kylar needs to pay 8 pairs * $8 per pair = $64. So, Kylar needs to pay $64 for the 16 glasses. The answer is 64.

Figure 8: Sample responses of a GSM8K question, from Phi-3.5 models, maximum generation length $L = 200$. The base model does not finish the generation, while the LaTRO model generates a short and correct answer.

**Question**: Water, ice, and steam all have different temperatures. What is the order from coldest to hottest?

**Options**: ['ice, water, steam', 'ice, steam, water','steam, ice, water','steam, water, ice']

**Response (Base model)**: To determine the order from coldest to hottest, let's evaluate each option.

- Ice: It is the solid form of water and is typically around 0°C (32°F).

- Water: It is the liquid form of water and is usually around 100°C (212°F) or 0°C (32°F) at standard pressure and temperature conditions, respectively.

- Steam: It is the gaseous form of water and is typically at a temperature equal to or greater than 100°C (212°F) at standard pressure conditions, depending on the pressure and other factors.

Option 1: 'ice, water, steam' - This order seems incorrect since steam is typically hotter than water at standard conditions.

Option 2: 'ice, steam, water' - This order is incorrect because ice is the coldest, but the relative order of water and steam is opposite of what we know.

Option 3:'steam, ice, water' - This order is incorrect since ice is colder than steam.

Option 4:'steam, water, ice' - This order seems the most plausible, considering that steam is hotter than water and water is hotter than ice.

The answer is 'steam, water, ice'.

**Response (SFT)**: water, ice, steam

**Response (LaTRO)**: To determine the order from coldest to hottest, let's analyze each option:

1. Ice has a solid form and a fixed temperature, which is typically around 0°C (32°F) at standard pressure.

2. Water is liquid and exists at a temperature of 0°C (32°F) at standard pressure, melting point of ice.

3. Steam is the gaseous state of water and is typically around 100°C (212°F) at standard pressure.

Based on this information, we can now order them from coldest to hottest:

First, we have ice, as it is the coldest. Then, we have water at 0°C, which is the middle temperature. Lastly, we have steam, which is the hottest.

So, the order from coldest to hottest is 'ice, water, steam'.

The answer is ['ice, water, steam'].

Figure 9: Sample responses of an ARC-Challenge question, from Llama-3.1 models, the errors are highlighted in red. The base model shows knowledge about basic physics but makes a logical error on the order.

