# OpenReview forum: "Language Models are Hidden Reasoners: Unlocking Latent Reasoning Capabilities via Self-Rewarding"
_ICLR.cc/2025/Conference — Submitted to ICLR 2025_

### Official Review · Reviewer_WHVo · 2024-10-25

**Soundness:** 2
**Presentation:** 3
**Contribution:** 2
**Rating:** 3
**Confidence:** 5

**Summary:**

This paper proposes a principled framework, LaTRO, to treat the reasoning process as sampling from latent distribution and enable LLMs themselves as reward models to evaluate the quality of reasoning rationales. The proposed LaTRO outperforms supervised fine-tuning baselines on 2 datasets.

**Strengths:**

1. The structure of this paper is clear, and it's easy to follow the main idea and the contribution.
2. The research problem this paper targets is very important to the community.
3. The motivation is sound and the benefits of the approach are clear.

**Weaknesses:**

1. Although modeling the rationales as latent variables to sample is well defined and proposed, the paper lacks a discussion with previously proposed reasoning methods that formulate the reasoning process as latent variables [1, 2] as well. Even though they are cited in the related work. Specifically, [1] proposes to sample diverse reasoning rationales to improve the prediction performance and also proposes an EM-like algorithm to improve the reward model and LLM alternatively. It would be great to have a discussion and comparison with these methods.
2. The paper does not compare with various prompting-based reasoning baselines mentioned in the related work section, such as tree-of-thought[3], and RAP[4], as well as fine-tuning baselines such as STaR[5], which is a missed opportunity to demonstrate its effectiveness. It would be better to compare them with metrics like training / inference computational cost and accuracy.
3. The paper does not provide a confidence interval, leading to unawareness of how the proposed LaTRO is robust to the initialization and randomness. It would be great to report the results from at least 3 times repetitions.
4. As the proposed LaTRO is a general approach, it would be great to evaluate it on more benchmark datasets to verify its effectiveness, such as HumanEval[6] and MBPP [7], which are popular in the current LLM reasoning community.


[1] Hu, Edward J., et al. "Amortizing intractable inference in large language models." arXiv preprint arXiv:2310.04363 (2023).

[2] Hoffman, Matthew Douglas, et al. "Training chain-of-thought via latent-variable inference." Advances in Neural Information Processing Systems 36 (2024).

[3] Yao, Shunyu, et al. "Tree of thoughts: Deliberate problem solving with large language models." Advances in Neural Information Processing Systems 36 (2024).

[4] Hao, Shibo, et al. "Reasoning with language model is planning with world model." arXiv preprint arXiv:2305.14992 (2023).

[5] Zelikman, Eric, et al. "Star: Bootstrapping reasoning with reasoning." Advances in Neural Information Processing Systems 35 (2022): 15476-15488.

[6] Chen, Mark, et al. "Evaluating large language models trained on code." arXiv preprint arXiv:2107.03374 (2021).

[7] Austin, Jacob, et al. "Program synthesis with large language models." arXiv preprint arXiv:2108.07732 (2021).

**Questions:**

See "Weaknesses".

---

### Official Review · Reviewer_XUrD · 2024-10-28

**Soundness:** 3
**Presentation:** 3
**Contribution:** 3
**Rating:** 5
**Confidence:** 3

**Summary:**

This paper introduces LaTRO, a framework that enhances LLM’s reasoning abilities by treating reasoning as sampling from a latent distribution and optimizing it with variational methods. LaTRO allows LLMs to improve their reasoning process and evaluation of reasoning quality simultaneously, without external feedback. Experiments on GSM8K and ARC-Challenge datasets demonstrate the effectiveness of LaTRO compared with the SFT training method.

**Strengths:**

1. LaTRO regards the reasoning process as sampling from a latent distribution and optimizes it using a variational method. This approach is different from prompt-based methods such as CoT and is closer to unsupervised learning. Besides, the feasibility of LaTRO is verified by mathematical proof.

2. This paper focuses on a very interesting topic, which enables an LLM to improve itself through a self-reward mechanism without external supervision and feedback signals.

**Weaknesses:**

1. Request a step-by-step description or flowchart of the LaTRO pipeline. Although a large number of formulas are used in this paper to explain each part，but it lacks a detailed description of the proposed method, which makes it difficult for readers to understand the complete pipeline of the proposed method and reproduce it.

2. As far as I know, there are some works to gradually enhance the capabilities of LLM (not limited to reasoning task), including prompt-based [1][2][3] and training-based methods [4][5][6], some of which do not use any feedback information to enhance the capabilities of LLM [7][8]. The author should discuss the differences between this work and these works and compare the performance of these works. It is necessary to add a dedicated subsection in the Related Work section discussing these specific works and their methods. If possible, include performance comparisons with these methods in the experimental section.

[1] When can llms actually correct their own mistakes? A critical survey of self-correction of llms

[2] Learning From Mistakes Makes LLM Better Reasoner

[3] Mirror: A Multiple-perspective Self-Reflection Method for Knowledge-rich Reasoning

[4] REFINER: Reasoning Feedback on Intermediate Representations

[5] CRYSTAL: Introspective Reasoners Reinforced with Self-Feedback

[6] SELF: Language-driven Self-evolution for Large Language model

[7] Small language modes can self-correct

[8] Think Thrice Before You Act: Progressive Thought Refinement in Large Language Models

3. In the experiments, the author only employs SFT training method and the base model as baselines, without comparing the performance with COT-SC mentioned in Section 3 and the self-improvement classic work mentioned above, making it difficult to demonstrate the advantages of the proposed LaTRO. So, please provide a more comprehensive analysis of how LaTRO performs relative to these additional baselines across different tasks and metrics.

**Questions:**

1. About self-evaluation task in LaTRO: how does LaTRO evaluate the probability of each reasoning path producing the correct answer? What does the conditional probability mention in Section 4.1 mean? Is there a task restriction for this evaluation method? What is the accuracy of self-evaluation? The authors did not discuss these questions in the paper, nor did they explore them in depth in the experiments.

2.  The authors only used a formula to explain the use of self-reward signals to achieve parameter updates, so what exactly does this update parameter refer to during the training phase? How is it implemented?  Suggest provide more detailed information.

---

### Official Review · Reviewer_dFSv · 2024-10-29

**Soundness:** 2
**Presentation:** 3
**Contribution:** 2
**Rating:** 3
**Confidence:** 4

**Summary:**

This paper introduces LaTRO, a novel approach that formulates Chain-of-Thought (CoT) reasoning as sampling from a latent distribution, optimized through variational techniques. By leveraging the probability of generating the correct answer as an implicit reward, LaTRO unifies the learning of both the policy and reward models, allowing large language models to refine reasoning paths in a self-rewarding manner. The authors demonstrate LaTRO’s effectiveness through experiments on the GSM8K and ARC-Challenge datasets across various model architectures. Their results indicate that latent reasoning capabilities within pre-trained language models can be unlocked and enhanced using this self-improvement framework.

**Strengths:**

1. The paper offers a novel perspective by framing reasoning as a process of sampling from a latent distribution and addressing it through variational methods.

2. The paper leverages the model's own probability estimates as an implicit reward, unifying the training of the policy and reward models.

3. The paper is well-organized and easy to follow.

**Weaknesses:**

1. The experimental setup lacks sufficient strong baselines, which are essential for a robust evaluation. Two key baselines to consider are:
  - A stronger SFT Baseline: Fine-tuning the policy model with correct reasoning paths. Given the availability of ground truth answers, multiple reasoning paths could be sampled, retaining only those that align with the ground truth. This baseline would provide a more rigorous comparison for evaluating LaTRO’s effectiveness in reasoning.
  - DPO Baseline: The authors could further fine-tune the policy model using the DPO algorithm, incorporating both correct and incorrect reasoning paths. Actually, the DPO algorithm aligns closely with LaTRO in its approach, as both methods aim to avoid training an explicit reward model. Including DPO as a baseline would highlight LaTRO’s strengths relative to an approach that similarly leverages implicit reward mechanisms.

2. The experimental scope is limited, as only two datasets and small models were tested.
 - Expanding the experiments to include a wider range of reasoning datasets would better assess the model's reasoning capabilities. Standard practice for evaluating reasoning in large language models includes diverse datasets that cover arithmetic reasoning (only GSM8K is not enough for arithmetic reasoning evaluation), commonsense reasoning, symbolic reasoning, and other reasoning types. Incorporating these would provide a more comprehensive evaluation.
 - Testing across varying model scales, especially with larger models, could provide insights into how the approach scales with model size and whether larger models yield better reasoning performance.

3. Although the authors claim that training did not rely on external feedback, the ground truth answers effectively serve as a form of implicit external feedback or reward.

**Questions:**

1. In Proposition 1, $p(y|x) = \int p(y|z,x) p(z|x) dz$ holds for any CoT-based method. Why, then, is CoT-SC introduced here?

2. What is the definition of "golden rationales", and why can’t ARC-Challenge have golden rationales?

3. What can the experimental results on ARC-Challenge demonstrate? The two baselines are too weak, as the SFT baseline did not utilize rationales.

---

### Official Review · Reviewer_n3ec · 2024-11-01

**Soundness:** 3
**Presentation:** 3
**Contribution:** 3
**Rating:** 5
**Confidence:** 2

**Summary:**

This paper proposesLaTRO, a novel framework aimed at enhancing the reasoning capabilities of LLMs. LaTRO addresses the challenge of improving reasoning during the training phase by formulating reasoning as sampling from a latent distribution and optimizing it through variational approaches. This method allows LLMs to self-improve their reasoning process and ability to evaluate the quality of reasoning without external feedback or reward models. The paper validates LaTRO's effectiveness through experiments on GSM8K and ARC-Challenge datasets, demonstrating significant improvements over base models and supervised fine-tuning approaches.

**Strengths:**

- The writing is clear and easy to follow.
- The discussed topic and motivation are both innovative and significant.

**Weaknesses:**

- Although I'm not familiar with the topic discussed in this article, I believe the experiments presented are too few and not comprehensive enough. Moreover, the datasets considered are limited to only GSM8K and ARC-Challenge, which lacks persuasiveness.
- The number of case study examples is too limited, with only a few instances in Figure 4 and the appendix, which is not convincing.
- The proposed method, LaTRO, especially the stability of the Self-reward component, has not been adequately considered.

**Questions:**

see weakness

---

### Official Review · Reviewer_b1ci · 2024-11-04

**Soundness:** 2
**Presentation:** 2
**Contribution:** 3
**Rating:** 3
**Confidence:** 4

**Summary:**

The work focuses on enhancing the reasoning abilities of large language models (LLMs) during the training phase without relying on external feedback. The authors motivate the work by raising an important question on improving the reasoning ability of LLMs during training phase since most prior works have focussed at achieving this at inference time. To do so, the authors propose to sample diverse reasoning rationales from latent distribution and optimize the LLM through a variational framework using the sampled rationales. The intuition behind application of the variational framework is well motivated through the objective of self-consistency based chain-of-thought prompting. Further, the work proposes to usethe  likelihood of generating correct answer conditioned on a rationale as a proxy to explicit reward models to optimise the LLM towards generating better reasoning chains. Results demonstrate that the proposed LaTRO method helps in improving the accuracy achieved for multiple LLMs such as Mistral-7B, Llama-3.1-8B etc. on GSM8K and ARC-Challenge datasets compared to the corresponding base and the SFT versions of the LLMs.

**Strengths:**

1. Enhancing the reasoning ability of LLMs without relying on feedback from any external LLM during the training phase is an important research question.
2. Suitability of applying the variational objective is well motivated, explained and justified in Sections 3 and 4.1.
3. Accuracy gains over the base and the SFT version of the LLMs are shown with some ablation studies on greedy decoding vs. self consistency based sampling. Further, the authors show that inference time scaling of number of reasoning samples obtained using LaTRO can additionally enhance the accuracy.

**Weaknesses:**

1. Presentation of the introduction section of the paper needs improvement - It should better motivate the need behind reducing the reliance on external LLMs/feedback for improving a given LLM. Further, it should provide information about why the variational framework is suitable and elaborate some details about the proposed method in the introduction itself.
2. Discussion on related work is very limited. Even though the proposed method is an effort to reduce reliance on external LLMs, it should still discuss and contrast against existing methods that leverage external LLMs (for example, [1,2]) as well as employ self-play based techniques to improve an LLM by itself with requiring help from any other LLM (for example, [3, 4]). The example references are only indicative (but not exhaustive) of the type of citations that should be included.
3. Lack of appropriate baselines - Even though the work claims and focusses at improving the base/SFT version of the LLM through the proposed LaTRO framework, it should still compare against some of the existing self-play trainable methods (by adapting them for 7B LLMs) as baselines (eg. [3] - on datasets where the ground-truth rationale is available and [4]). Such methods improve the rationales generated by an LLM by exploring the generation space and discriminating better rationales from the inferior ones.
4. More reasoning datasets such as CSQA, Hellaswag etc. should be considered for evaluation studies.

[1] Distilling Step-by-Step! Outperforming Larger Language Models with Less Training Data and Smaller Model Sizes. Cheng-Yu Hsieh, Chun-Liang Li, Chih-kuan Yeh, Hootan Nakhost, Yasuhisa Fujii, Alex Ratner, Ranjay Krishna, Chen-Yu Lee, and Tomas Pfister. In Findings of the Association for Computational Linguistics: ACL 2023, pages 8003–8017, Toronto, Canada. Association for Computational Linguistics.

[2] Zephyr: Direct Distillation of LM Alignment. Lewis Tunstall and Edward Emanuel Beeching and Nathan Lambert and Nazneen Rajani and Kashif Rasul and Younes Belkada and Shengyi Huang and Leandro Von Werra and Clementine Fourrier and Nathan Habib and Nathan Sarrazin and Omar Sanseviero and Alexander M Rush and Thomas Wolf. First Conference on Language Modeling, 2024.

[3] Self-Play Fine-Tuning Converts Weak Language Models to Strong Language Models. Zixiang Chen and Yihe Deng and Huizhuo Yuan and Kaixuan Ji and Quanquan Gu. ICML 2024.

[4] Self-Rewarding Language Models. Weizhe Yuan, Richard Yuanzhe Pang, Kyunghyun Cho, Xian Li, Sainbayar Sukhbaatar, Jing Xu, Jason Weston

**Questions:**

1. Line 290: It is not clear about how the first gradient term in Eq. 4 would lead to optimising the LLM policy to generate higher quality rationales. Further elaboration is needed as to why generating/optimising the likelihood of input question/instruction conditioned on the rationale would lead to better reasoning.
2. Lines 414-415: Please support with examples about why improvements on ARC-Challenge are relatively lesser. The magnitude of improvements is not at all an issue, however, it should be better demonstrated that why better reasoning chains are not leading to higher improvements. Does this make ARC-Challenge ill-suited for this study since it involves limited reasoning scope?
3. Line 430: Why is it needed to generate rationales that are as long as 500 tokens? It would be good to show the usefulness of the information contained in the longer rationales and why does it lead to better performance before saturating.

---

### Author Response · Authors · 2024-11-27

We thank the reviewers for their insightful advices. It is good hear that our method is presented in a clear way and we also agree that we did not conduct enough experiments to demonstrate the effectiveness of our algorithm empirically.

Due to recent lack of computing resources we could not experiment with additional baselines and datasets at this point. We finished some experiments but we feel it would be more comprehensive to release the revision when we have more results.

---

### Meta-Review · Area_Chair_45Cp · 2024-12-22

**Metareview:**

This paper aims to enhance the reasoning abilities of LLMs at the stage of training. It formulates the reasoning process as sampling from latent distribution and optimizes the LLM through a variational framework using the sampled rationales.  While the target problem is important and the method is well motivated, main concerns of the reviewers include:  1. discussion on related work is limited; 2. the experiment did not compare with strong baselines,  and only two datasets and small models were tested.    The authors did not provide formal rebuttal, and instead acknowledge that they "did not conduct enough experiments to demonstrate the effectiveness of our algorithm empirically". AC agrees and suggests the authors revise and improve the submission accordingly.

**Additional Comments On Reviewer Discussion:**

Two main concerns from the reviewers:

1. discussion on related work is limited;
2. the experiment did not compare with strong baselines,  and only two datasets and small models were tested.

No formal rebuttal is provided. Instead, the authors acknowledged "did not conduct enough experiments to demonstrate the effectiveness of our algorithm empirically".

I tend to agree with the reviewers that this work requires a more comprehensive and convincing evaluation.

---

### Decision · Program_Chairs · 2025-01-22

Reject